# Factors Associated with Withdrawal Time in European Colonoscopy Practice: Findings of the European Colonoscopy Quality Investigation (ECQI) Group

**DOI:** 10.3390/diagnostics12020503

**Published:** 2022-02-15

**Authors:** Cristiano Spada, Anastasios Koulaouzidis, Cesare Hassan, Pedro Amaro, Anurag Agrawal, Lene Brink, Wolfgang Fischbach, Matthias Hünger, Rodrigo Jover, Urpo Kinnunen, Akiko Ono, Árpád Patai, Silvia Pecere, Lucio Petruzziello, Jürgen F. Riemann, Harry Staines, Ann L. Stringer, Ervin Toth, Giulio Antonelli, Lorenzo Fuccio

**Affiliations:** 1Digestive Endoscopy Unit and Gastroenterology, Fondazione Poliambulanza, 25124 Brescia, Italy; 2Digestive Endoscopy Unit, Università Cattolica del Sacro Cuore, 00168 Rome, Italy; 3Department of Medicine, OUH Svendborg Sygehus, 5700 Svendborg, Denmark; akoulaouzidis@hotmail.com; 4Department of Clinical Research, University of Southern Denmark (SDU), 5000 Odense, Denmark; 5Surgical Research Unit, OUH, 5000 Odense, Denmark; 6Department of Social Medicine and Public Health, Pomeranian Medical University, 70-204 Szczecin, Poland; 7Endoscopy Unit, IRCCS Humanitas Clinical and Research Center, 20089 Milan, Italy; cesareh@hotmail.com; 8Gastroenterology Department, Centro Hospitalar e Universitário de Coimbra, 3000-075 Coimbra, Portugal; pedro.amaro1967@gmail.com; 9Gastroenterology, Doncaster Royal Infirmary, Doncaster DN2 5LT, UK; anurag.agrawal1@nhs.net; 10Gastro Unit, Division of Endoscopy, Herlev and Gentofte Hospital, Copenhagen University, 2730 Herlev, Denmark; lene.brink@regionh.dk; 11Gastroenterologie und Innere Medizin, 63739 Aschaffenburg, Germany; wuk.fischbach@gmail.com; 12Independent Researcher for Internal Medicine, 97070 Würzburg, Germany; mhuenger@gmx.net; 13Instituto de Investigación Sanitaria ISABIAL—Servicio de Medicina Digestiva, Hospital General Universitario de Alicante, 03010 Alicante, Spain; rodrigojover@gmail.com; 14Department of Gastroenterology, Tampere University Hospital, 33521 Tampere, Finland; urpo.kinnunen@pshp.fi; 15Department of Gastroenterology, Hospital Clínico Universitario Virgen de la Arrixaca, El Palmar, 30120 Murcia, Spain; ono.akiko@gmail.com; 16Department of Gastroenterology and Medicine, Markusovszky University Teaching Hospital, 9700 Szombathely, Hungary; pataiarpaddr@gmail.com; 17Digestive Endoscopy Unit, Fondazione Policlinico Universitario A. Gemelli IRCCS, 00168 Rome, Italy; silvia.pecere@gmail.com (S.P.); luciopetruzziello@gmail.com (L.P.); 18Department of Medicine C, Klinikum Ludwigshafen, 67063 Ludwigshafen, Germany; riemannj@garps.de; 19LebensBlicke Foundation, 67063 Ludwigshafen, Germany; 20Sigma Statistical Services Ltd., Saint Andrews KY16 0BD, UK; harry.j.staines@gmail.com; 21ECQI Secretariat, Buckinghamshire HP17 8ET, UK; ann.stringer@aspenmedicalmedia.com; 22Department of Gastroenterology, Skåne University Hospital, Lund University, 205 02 Malmö, Sweden; ervin.toth@med.lu.se; 23Department of Anatomical, Histological, Forensic Medicine and Orthopedics Sciences, “Sapienza” University of Rome, 00185 Rome, Italy; giulio.antonelli@gmail.com; 24Gastroenterology and Digestive Endoscopy Unit, Ospedale dei Castelli, Ariccia, 00040 Rome, Italy; 25Gastroenterology Unit, Department of Medical and Surgical Sciences, S. Orsola-Malpighi Hospital, 40138 Bologna, Italy; lorenzofuccio@gmail.com

**Keywords:** colonoscopy, colonoscopy standards, withdrawal time, quality measures

## Abstract

The European Colonoscopy Quality Investigation (ECQI) Group aims to raise awareness for improvement in colonoscopy standards across Europe. We analyzed data collected on a sample of procedures conducted across Europe to evaluate the achievement of the European Society of Gastrointestinal Endoscopy (ESGE) mean withdrawal time (WT) target. We also investigated factors associated with WT, in the hope of establishing areas that could lead to a quality improvement. Methods: 6445 form completions from 12 countries between 2 June 2016 and 30 April 2018 were considered for this analysis. We performed an exploratory analysis looking at WT according to the ESGE definition. Stepwise multivariable logistic regression analysis was conducted to determine the most influential associated factors after adjusting for the other pre-specified variables. Results: In 1150 qualifying colonoscopies, the mean WT was 7.8 min. Stepwise analysis, including 587 procedures where all inputs were known, found that the variables most associated with mean WT were a previous total colonoscopy in the last five years (*p* = 0.0011) and the time of day the colonoscopy was performed (*p* = 0.0192). The main factor associated with a WT < 6 min was the time of day that a colonoscopy was performed. Use of sedation was the main factor associated with a higher proportion of WT > 10 min, along with a previous colonoscopy. Conclusions: On average, the sample of European practice captured by the ECQI survey met the minimum standard set by the ESGE. However, there was variation and potential for improvement.

## 1. Introduction

The adenoma detection rate (ADR) is a validated quality measure that colonoscopists should constantly seek to improve. While ADR may be considered the primary quality indicator, it is a function of other quality measures, such as cecal intubation rate, withdrawal time (WT), and quality of bowel preparation. Colonoscope WT is considered a surrogate measure for the time spent investigating the mucosa to identify pathology [1].

The European Colonoscopy Quality Investigation (ECQI) Group (www.ecqigroup.org (accessed on 13 January 2022) comprises specialists and advisors and aims to raise awareness of the need for improvement in colonoscopy standards across Europe. The ECQI is a collaborative working party seeking cooperation and input from all stakeholders in the field of colonoscopy. The ECQI’s aim is not to create new quality criteria, but rather to document dissemination of the European Society of Gastrointestinal Endoscopy (ESGE) guidelines and record their implementation in daily practice throughout Europe [2]. At the inaugural meeting of the ECQI Group, convened in 2013, to discuss quality in colonoscopy, the Group recommended devising a clinical practice questionnaire to evaluate the current practice of endoscopists across Europe. The main findings from this questionnaire have already been published [2].

In 2017, the ESGE published performance measures for lower gastrointestinal endoscopy [1]. The ESGE recommend that endoscopy services across Europe adopt the key performance measures for measurement and evaluation in daily practice at a center and endoscopist level. As a secondary analysis, we analyzed a sample of procedures conducted across Europe, between June 2016 and April 2018, in order to evaluate the current achievement of WT standards, as defined by the ESGE. We also analyzed data collected on procedures with regard to factors associated with WT, in the hope of establishing areas that could lead to an improvement in quality [1].

## 2. Methods

### 2.1. Questionnaire Development

The questionnaire was developed with consideration of the ESGE quality standards [3]. An iterative process was used to hone the questionnaire, ensuring that the time to complete the form was not too onerous [4]. This analysis uses the version finalized in 2016 and available for completion from 2 June 2016 (see Appendix A).

### 2.2. Recruitment

Participation was open to all Europe-based colonoscopists via web-based registration on the ECQI Group website. Awareness of the questionnaire came from abstracts, posters, presentations at national and international congresses, and individual communications from ECQI Group members. Interested participants applied via the ECQI Group website or to the ECQI Group Secretariat. Following verification, log-in access to the web-based questionnaire site was provided by email.

### 2.3. Ethics

This survey was performed with anonymized data, collected during regular clinical care, representing an audit of routine endoscopic practice against quality standards. Accordingly, participating physicians were encouraged to follow relevant local regulations. Furthermore, contemporary (pre-General Data Protection Regulation, GDPR, 2018) guidance was followed at the time of data collection.

### 2.4. Dataset

Form completions from 2 June 2016 to 30 April 2018 were included in this analysis.

### 2.5. Withdrawal Time

The ESGE define the WT as time spent on withdrawal of the endoscope from the cecum to the anal canal and inspection of the entire bowel mucosa at negative (no biopsy or therapy) screening or diagnostic colonoscopy [1]. We identified screening and diagnostic colonoscopies in our dataset using the reason for procedure question. ‘Signs and symptoms’ was classified as diagnosis, and the responses, ‘Screening due to familial risk’, ‘Screening without pre-screening test’ and ‘Following positive screening test’ were classified as screening. The ‘Other’ response free-text section was reviewed and responses re-classified as the above, as appropriate. All responses that were neither diagnosis nor screening were excluded from this analysis group. We used the ‘Retraction time’ recorded in colonoscopies reporting the cecum as the intended endpoint, that reported reaching the intended endpoint, and did not report any endoscopic intervention, in order to provide a mean WT.

As well as mean WT, analysis was performed using the cut-off points of 6 min and 10 min, as per the ESGE recommendations for minimum and target mean WT, respectively [1]. Descriptive statistics were provided for each variable category, and one-way analysis of variance was used to test for equality of mean across variable categories. WT < 6 min and WT > 10 min were treated as separate binary responses.

### 2.6. Statistical Analysis

To preserve anonymity, only the patient’s year of birth was recorded. Age at the date of the procedure was derived assuming the date of birth was 30 June. Quantitative variables are presented as mean ± standard deviation (SD). Binary responses are presented as frequency and percentage.

Univariate binary logistic regression models were used to determine the association of individual variables with an endpoint using pre-defined categories. For selected analyses involving pairs of variables, the interaction term was added to the model, in addition to the two main effects. Stepwise multivariable logistic regression analysis was conducted to determine the most influential associated factors after adjusting for the other pre-specified variables. Stepwise analysis was performed on the following variables: age < 50; gender; BMI categories; inpatient status; reason for procedure; time of colonoscopy; previous total colonoscopy in the last five years; Boston Bowel Preparation Score (BBPS) of adequate [5]; and sedation used. A variable was included in the stepwise model if the p value for entering the model was <0.05, and removed if the *p* value was >0.10. Such analysis was restricted to the set of procedures for which all the pre-specified variables are known.

No adjustment for multiplicity was made with a *p* value < 0.05 used to define significance. All analyses were performed using the statistical software package SAS version 9.4. (Cary, NC, USA). As sensitivity analyses, missing data were imputed using multiple imputation by chained equations (MICE) [6]. A total of 100 multiple imputation (MI) datasets were created imputing missing data for individual variables using the other potential variables selected for the stepwise model. Analyses were conducted separately for each MI dataset, and the results for effects pooled using Rubin’s method [7]. The mean *p* value for the overall model was also calculated.

## 3. Results

A total of 6445 completed procedure forms from 12 countries were considered for inclusion in analysis. Forms were received from 25 academic hospitals (n = 2270), 14 hospitals (n = 1235), 8 private institutions (n = 2657), 3 group practices (n = 160), and 1 other center (n = 123). There were 1150 procedures that met the criteria for the WT analysis (see Figure 1). These were divided between 41 institutions, with a median of 10 procedures per institution (range 1–154).

### 3.1. Withdrawal Time

The overall mean WT was 7.8 ± 3.1 min, and the median WT was 7 min. The proportion of patients receiving a WT less than 6 min was 13.0% (150/1150), and the proportion with a WT greater than 10 min was 10.8% (124/1150). Stepwise analysis, including 587 procedures where all inputs were known, found that the variables most associated with mean WT were a previous total colonoscopy in the last five years (yes 8.9 ± 3.9 vs. no 7.8 ± 3.2; p = 0.0011) and the time of day the colonoscopy was performed (8.3 ± 2.7, 7.9 ± 4.0 and 6.6 ± 2.2, respectively, for patients whose colonoscopy was performed between 07:00–11:59, 12:00–17:59, and 18:00–19:59; *p* = 0.0192). Following MI, mean *p* values for variables remained broadly similar. with the notable exception of the time of day that the colonoscopy was performed, which lost significance. Stepwise analysis, performed on the 100 MICE datasets, showed adequate cleansing, sedation, patient type, and reason for procedure were included in each of the 100 models. The time of day was included in 76 models, and total colonoscopy within the last five years was included in 10 models.

WT was not influenced by either age or BMI (data not shown). The factors significantly associated with mean WT are shown in Table 1. Gender had no significant influence on mean WT or the proportion with a WT over 10 min; however, men were significantly less likely to receive a WT of less than 6 min (10.3% vs. 14.9%; OR 0.657, 95%CI 0.456, 0.944; *p* = 0.023). The only other factor with a significant association to the proportion of patients with a WT less than 6 min was the time of day the colonoscopy was performed (Figure 2), which was the only associated factor in the stepwise analysis. In the stepwise analysis, the proportion of patients with a withdrawal time less than 6 min was 12.3%, 19.7%, and 35.7%, respectively, for a colonoscopy performed between 07:00–11:59, 12:00–17:59, and 18:00–19:59. The OR (vs. colonoscopies between 07:00 and 11:59) were 1.751 (95%CI 1.097, 2.796; *p* = 0.0189) and 3.968 (95%CI 1.925, 8.181; *p* = 0.0002), respectively, for colonoscopies performed between 12:00–17:59 and 18:00–19:59. Following MI, the mean *p* value for the time of day increased to *p* = 0.002, while all other variables remained non-significant, except gender, which remained at *p* = 0.023, as there were no missing data for this variable. Stepwise analysis following MI retained the time of day, but gender was also included in all 100 models.

Stepwise analysis of the variables associated with the proportion of patients receiving a WT greater than 10 min showed that sedation was the primary factor (yes 14.3% vs. no 7.3%; OR 2.115, 95%CI 1.083, 4.134; *p* = 0.0284), followed by a previous total colonoscopy in the last five years (yes 18.4% vs. no 11.2%; OR 1.805, 95%CI 1.018, 3.201; *p* = 0.0478). The influence of individual variables is shown in Table 2. Following MI, sedation use remained in all 100 stepwise models; however, it was joined in all models by patient type, reason for procedure, and adequate cleansing. Total colonoscopy in the previous five years only appeared in one of the 100 MI models.

The reason for performing a screening procedure had an effect on WT (*p* = 0.005) with ‘Following positive screening test’ associated with a longer mean WT (9.3 ± 3.3 min) than either ‘Screening due to familial risk’ (7.9 ± 3.0 min) or ‘Screening without pre-screening test’ (8.1 ± 4.4 min). This difference was also apparent in the proportions receiving WTs of less than 6 min (4.3% vs. 17.5% and 16.2%, respectively, *p* = 0.008), and the trend was apparent, but not statistically significant, in the proportion greater than 10 min (19.8% vs. 11.9% and 14.9%, respectively, *p* = 0.232).

We performed analysis of the effect of an abnormal endoscopic finding on WT to examine whether the reason for short WTs was the discovery of pathology (noting that procedures involving endoscopic intervention were excluded from WT analysis). Abnormal endoscopic finding had no association with mean WT: yes 8.0 ± 3.5 min (n = 461) vs. no 7.7 ± 2.8 min (n = 681) (*p* = 0.112). There was also no effect on the proportion of patients with a WT less than 6 min: yes 14.1% (65/461) vs. no 12.5% (85/681) (*p* = 0.427). However, there was an increase in the proportion of those with a WT over 10 min for those with an abnormal endoscopic finding: yes 12.8% (59/461) vs. no 9.0% (61/681); OR 1.492, 95%CI 1.021, 2.180 (*p* = 0.039). Abnormal endoscopic findings included cancer (n = 19), diverticula (n = 322), inflammation (n = 46), polyps (n = 52), hemorrhoids (n = 44), and other findings (n = 40).

An analysis was conducted to see if any particular variable interacted with the effect of time of day the colonoscopy was performed on the length of WT. This analysis was restricted to 850 procedures, with responses to ‘Time of colonoscopy’ that could not clearly be assigned to a time of day category being excluded. The reason for procedure (diagnostic vs. screening) was the only variable found to significantly interact with the WT, both mean WT (*p* = 0.005) and the proportion with a WT under 6 min (*p* = 0.015), but not the proportion over 10 min (*p* = 0.420). See Figure 3 for how time colonoscopy was performed influences the proportion with a WT of less than 6 min according to the reason for the procedure.

### 3.2. Impact of Sedation

The type of sedation used had a significant association with mean WT (*p* < 0.001) (Figure 4). In particular, the proportion of patients with a WT of >10 min was substantially greater with midazolam used in combination with propofol, and to a lesser extent, but still significant, with the use of midazolam and opioid together, nitrous oxide alone, and propofol alone (Table 3).

## 4. Discussion

The ESGE recommends that the mean WT is at least 6 min with a target standard of 10 min [1]. We found a mean WT in qualifying colonoscopies of 7.8 min. A WT over 10 min occurred in only 10.8% of procedures, while a WT under 6 min occurred in 13.0% of procedures.

In analysis of known data, the main factors associated with mean WT were whether the patient had a previous colonoscopy within the past five years and the time of day that the colonoscopy was performed. The main factor associated with a WT of less than 6 min was the time of day the colonoscopy was performed. The use of sedation was the main factor associated with a higher proportion of WTs over 10 min, along with a previous colonoscopy.

Our known data analysis found a significant association between the time of day that a colonoscopy is performed and mean WT, with lower times later in the day, culminating in an increased proportion with a WT less than 6 min. This finding is in agreement with Marcondes et al., who found a roughly 20% reduction in WT from the first to last colonoscopy of the day [8]. A decline in ADR and PDR was also apparent in colonoscopies performed later in the clinic schedule. Teng et al. also found a reduction in WT from morning to afternoon sessions. Indeed, along with increasing patient age, WT was found to be a significant predictor of decline in ADR from morning to afternoon sessions. Furthermore, after controlling for WT and advancing age, the difference in ADR between morning and afternoon was no longer significant. [9] Both papers conclude that the most likely reason for the reduction in WT is rushing procedures later in the day [8,9].

The ESGE states that colonoscopy needs adequate time allocated for the entire procedure (including discussion with the patient, sedation, insertion, withdrawal, and therapy) [1]. Time pressure due to inadequate time slots may impair colonoscopy quality. A minimum standard of 30 min for a clinical and primary screening colonoscopy, and 45 min for a colonoscopy following positive fecal occult blood testing is recommended. Our findings of an association between time of day and WT could be indicative of insufficient time allotted to procedures leading to ‘rushing’ of procedures towards the end of clinic time.

Further to the findings of Singh et al. that a difference in ADR between morning and afternoon colonoscopies was mainly found in female patients [10], we looked at whether any of our variables interacted with the relationship between WT and the time that the colonoscopy was performed. We found no effect of gender on WT. However, we found that the factor significantly associated with the increase in proportion of patients with a WT <6 min according to time of day was screening being the reason for procedure. This suggests that screening patients, in particular, are more likely to receive shorter WTs later in the day.

Evidently, shortening WTs in screening patients can have negative consequences. Kumar et al. showed that a WT of 3 min was associated with a substantial increase in the adenoma miss rate compared with a WT of 6 min (48.0% vs. 22.9%, *p* = 0.0001) [11]. When identifying incident interval cancers via cancer registry linkages that were diagnosed within 5.5 years after the screening examination, Shaukat et al. found that physicians’ mean annual WTs were inversely associated with cancer incidence (*p* < 0.0001). Compared with WTs ≥ 6 min, the adjusted incidence rate ratio for WTs of <6 min was 2.3 (95% CI: 1.5, 3.4; *p* < 0.0001) [12].

The ESGE recommend the measurement of WT when ADR is below the minimum standard of 25% [1]. Our findings indicate that WT is an important factor in screening and diagnostic colonoscopies. It is also a factor that is potentially under colonoscopist control [13]. Encouraging longer WTs by allowing sufficient appointment time, and possibly using interventions to encourage achievement of higher mean WTs, has the potential to improve polyp detection [14,15,16,17,18,19]. Indeed, simply being aware that WT is being monitored has been shown to provide improvement [20,21,22].

The use of sedation was significantly associated with mean WT and was a key factor associated with WTs >10 min. An international expert panel agreed that patients undergoing colonoscopy should be offered the choice of sedation or no sedation [23]. The majority of procedures in our dataset were performed with some form of sedation administered: 71.5% of the WT analysis.

Propofol for sedation during colonoscopy for generally healthy individuals can lead to faster recovery and discharge times, as well as increased patient satisfaction without an increase in complications [24]. However, there is still debate over whether it is appropriate for propofol to be administered without the presence of an anesthesiologist [25,26].

Analyses have demonstrated that administration of propofol by an endoscopist or specialist nurse is safe and effective [27,28,29]. Patient satisfaction has also been shown to be better with the use of propofol compared with a combination of midazolam and opioid [28,30]. However, the use of propofol vs. midazolam plus opioid does not necessarily lead to an increase in ADR/PDR or WT [31,32,33].

### 4.1. Multiple Imputation Analysis

MI was used as a sensitivity analysis. The main causes of missing data for the stepwise models were the time of day that the colonoscopy was performed and the BMI category. The proportions for the observed and imputed data for each of these variables were reasonably matched. The univariate results using the MI and observed data were broadly in agreement, although the significance of the time of day that the colonoscopy was performed was lost for WT itself. This could reflect an inherent problem with imputing the time of day the colonoscopy was performed, since these data may not be missing at random. For example, some clinics might only perform colonoscopies in the morning.

With the MI data, differences were found for the stepwise models. First, gender in addition to time of day was included in each of the 100 stepwise models for the proportion of procedures with WT of less than 6 min. Second, for the proportion of procedures with WT greater than 10 min, patient type, reason for procedure, and adequate cleansing, rather than a total colonoscopy within the last five years, were the most important factors together with sedation (each was included in each of the 100 models). Third, for the WT, adequate cleansing, sedation, patient type, and reason for procedure were included in each of the 100 models, while time of day and total colonoscopy within the last five years were included in 76 and 10 models, respectively.

Thus, while no clarity was provided by MI on the most important factors influencing WT overall, the time of day that the colonoscopy was performed and sedation use were confirmed as highly influential factors for WT less than 6 min and greater than 10 min, respectively.

### 4.2. Limitations

The publication of the ESGE performance measures occurred after this version of the questionnaire was compiled, therefore there are some areas where our measures do not exactly match those specified by the ESGE. Some issues with the completion of the questionnaire were also identified, which have led to some responses being excluded from analysis due to either incorrect interpretation of the question or implausible responses. This has restricted the number of procedures that could be included in the multivariable analyses. In future, with a revised questionnaire design, we should be able to substantially reduce the number of non-eligible responses, which should improve the power of the analysis.

The institutions and practitioners completing this questionnaire are from a range of countries across Europe. While this is a strength in that it provides a variety of practices enabling the assessment of a wide range of variables to see which influence outcomes, it also means that the practices in some countries may be overrepresented and could skew results. Selection bias is also possible, as we have no control over which procedures practitioners choose to document. Additional variables that could have influenced the results include the day of the week that the colonoscopy was performed and the level of experience of the colonoscopist. These variables could be included in future questionnaires/analyses.

This analysis was exploratory in nature and guided by findings from these data. The observations would ideally be confirmed by prospective studies, for example by using training and validation subsets.

## 5. Conclusions

On average, the sample of European practice captured by the ECQI survey meets the minimum standards set by the ESGE. However, there is variation and potential for improvement. In particular, educational initiatives regarding the importance of maintaining WTs throughout the day, particularly in screening patients, could yield benefits in terms of cancer detection. The use of sedation has the potential to improve WTs and, in accordance with local protocols, should be offered to all patients undergoing colonoscopy.

## Figures and Tables

**Figure 1 diagnostics-12-00503-f001:**
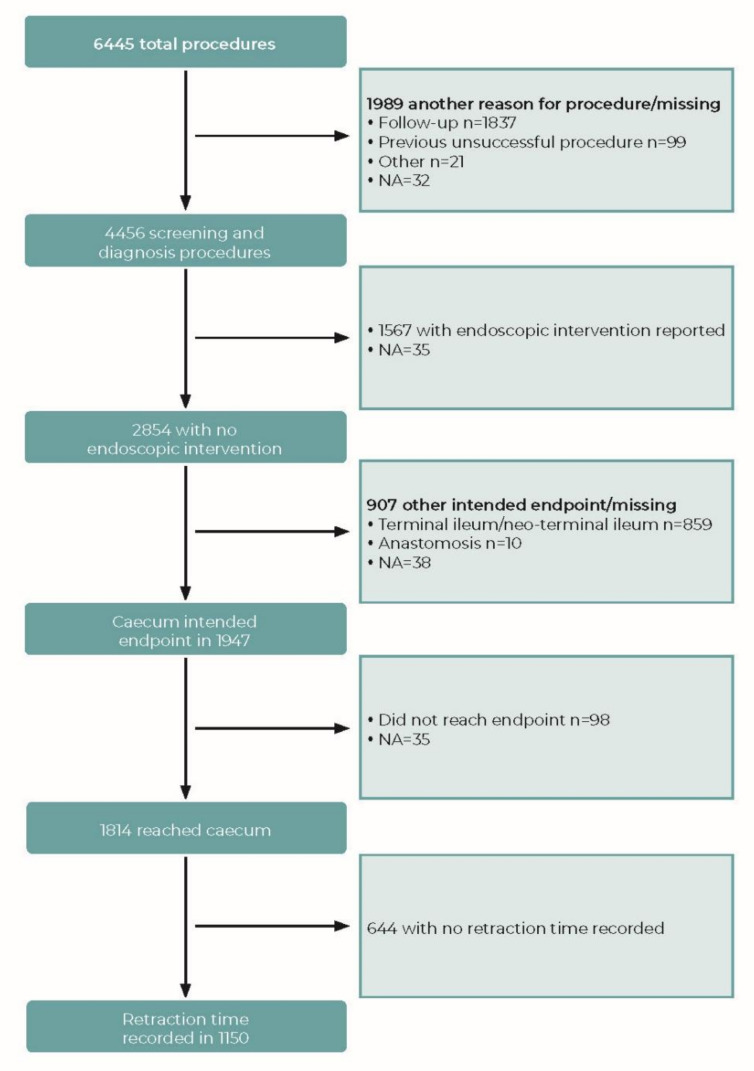
Flowchart indicating data inclusion/exclusion for analysis of withdrawal time (WT). NA: not answered.

**Figure 2 diagnostics-12-00503-f002:**
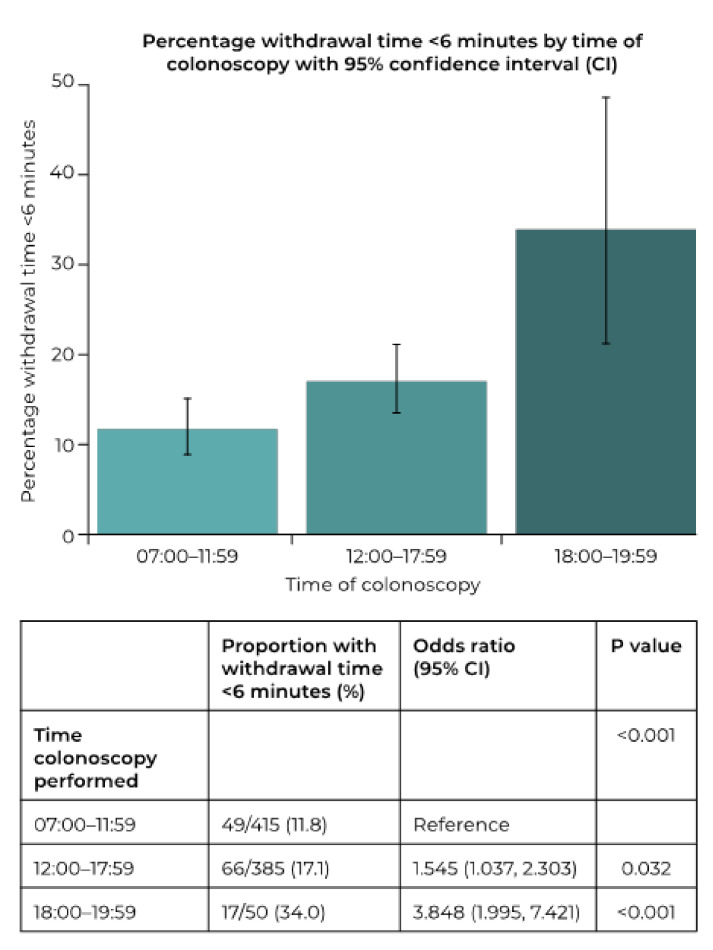
Influence of the time of day that a procedure is performed upon the proportion of procedures with a withdrawal time under 6 min.

**Figure 3 diagnostics-12-00503-f003:**
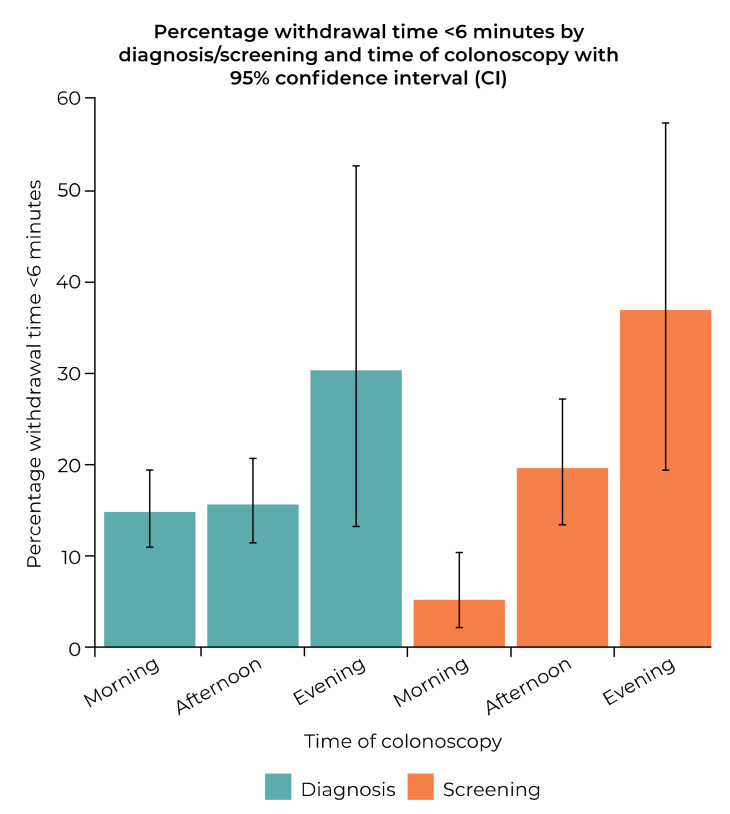
Influence of time colonoscopy was performed on the proportion of procedures with withdrawal time < 6 min according to reason for procedure being diagnosis or screening.

**Figure 4 diagnostics-12-00503-f004:**
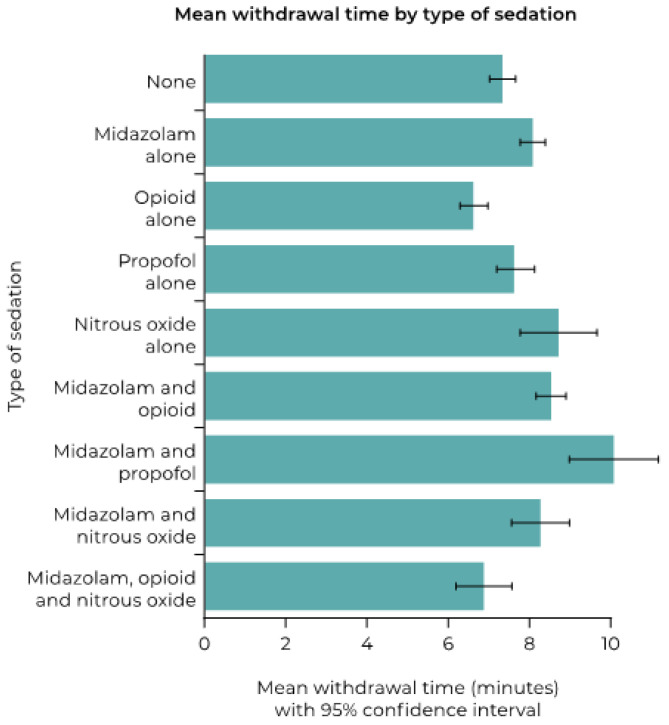
Mean withdrawal time (minutes) according to type of sedation used.

**Table 1 diagnostics-12-00503-t001:** Influence of variables on mean withdrawal time (minutes).

Variable	Number	Mean ± SD	95%CI	*p* Value for Variable	Mean *p* Value Following MI
**Patient type**				**<0.001**	**0.001**
Outpatient	942	7.7 ± 3.0	7.5, 7.9		
Inpatient	98	9.0 ± 4.3	8.2, 9.9		
Missing	110				
**Reason for procedure**				**<0.001**	**NA**
Diagnosis	754	7.5 ± 2.7	7.3, 7.7		
Screening	396	8.4 ± 3.7	8.1, 8.8		
Missing	0				
**BBPS**				**0.004**	**0.004**
BBPS < 6	142	8.5 ± 3.7	7.9, 9.2		
BBPS ≥ 6	998	7.7 ± 3.0	7.5, 7.9		
Missing	10				
**Time colonoscopy performed**				**0.015**	**0.098**
07:00–11:59	415	7.8 ± 2.6	7.5, 8.0		
12:00–17:59	385	7.4 ± 3.5	7.1, 7.8		
18:00–19:59	50	6.5 ± 2.0	5.9, 7.1		
Missing	300				
**Previous total colonoscopy in last 5 years**				**0.018**	**0.020**
No	875	7.7 ±3.1	7.5, 7.9		
Yes	168	8.3 ± 3.5	7.8, 8.9		
Missing	107				
**Sedation used**				**<0.001**	**<0.001**
No	326	7.3 ± 3.1	6.9, 7.6		
Yes	816	8.1 ± 3.1	7.8, 8.3		
Missing	8				

**Table 2 diagnostics-12-00503-t002:** Influence of individual variables on the proportion of patients with a withdrawal time greater than 10 min.

Variable	Proportion with Withdrawal Time > 10 min (%)	Odds Ratio (95%CI)	*p* Value	*p* Value for Variable	Mean *p* Value Following MI
**Patient type**				**0.005**	**0.021**
Outpatient	94/942 (10.0)	Reference			
Inpatient	19/98 (19.4)	2.170 (1.259, 3.739)	0.005		
Missing	110				
**Reason for procedure**				**<0.001**	**NA**
Diagnosis	63/754 (8.4)	Reference			
Screening	61/396 (15.4)	1.997 (1.372, 2.907)	<0.001		
Missing	0				
**BBPS**				**0.007**	**0.007**
BBPS < 6	25/142 (17.6)	Reference			
BBPS ≥ 6	99/998 (9.9)	0.515 (0.319, 0.832)	0.007		
Missing	10				
**Time colonoscopy performed**				**0.479**	**0.561**
07:00–11:59	44/415 (10.6)	Reference			
12:00–17:59	34/385 (8.8)	0.817 (0.510, 1.308)	0.399		
18:00–19:59	3/50 (6.0)	0.538 (0.161, 1.802)	0.315		
Missing	300				
**Previous total colonoscopy in last five years**				**0.083**	**0.104**
No	90/875 (10.3)	Reference			
Yes	25/168 (14.9)	1.525 (0.946, 2.458)	0.083		
Missing	107				
**Sedation used**				**0.002**	**0.002**
No	20/326 (6.1)	Reference			
Yes	103/816 (12.6)	2.210 (1.344, 3.634)	0.002		
Missing	8				

**Table 3 diagnostics-12-00503-t003:** Proportion of procedures with a withdrawal time greater than 10 min according to type of sedation (combinations with a frequency <10 were excluded from analysis).

	Proportion with Withdrawal Time >10 min (%)	Odds Ratio (95%CI)	*p* Value
Type of sedation			<0.001
None	20/326 (6.1)	Reference	
Midazolam alone	2/100 (2.0)	0.312 (0.072, 1.360)	0.121
Opioid alone	2/64 (3.1)	0.494 (0.112, 2.166)	0.349
Propofol alone	24/218 (11.0)	1.893 (1.018, 3.519)	0.044
Nitrous oxide alone	6/40 (15.0)	2.700 (1.015, 7.185)	0.047
Midazolam and opioid	55/323 (17.0)	3.140 (1.835, 5.374)	<0.001
Midazolam and propofol	12/34 (35.3)	8.345 (3.616, 19.259)	<0.001
Midazolam and nitrous oxide	0/11 (0)	0	–
Midazolam, opioid, and nitrous oxide	0/15 (0)	0	–

## Data Availability

An application form to access the data is available from the ECQI Secretariat upon receipt of a rationale and statistical analysis plan.

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
