# Peer review of "Factors Associated with Withdrawal Time in European Colonoscopy Practice: Findings of the European Colonoscopy Quality Investigation (ECQI) Group"

_diagnostics, 2022, doi:10.3390/diagnostics12020503_

Round 1

Reviewer 1 Report

Thank you for edits and additions.

Author Response

No further modifications are needed

Reviewer 2 Report

The manuscript is significantly improved from the original version despite the significant data gaps addressed in 4.1 (Multiple imputations analysis).

Minor issues:

  1. The reviewer remains concerned about asymmetric numbers of procedures affecting results. For instance, if a single institution accounted for >50% of the patients in this series, the results may not be generalizable. The authors chose not to address this issue on my initial review. Perhaps they could do so in a supplementary table giving general numbers (e.g., <100 pts, >100, >500 per site, etc.). This would allow the reader some insight into whether one or two institutions provided the bulk of the data.
  2. In the Introduction, please state the bulk of the findings in this patient group have already been published (Reference 2) and that this is a secondary analysis looking at colonoscope withdrawal time.

Author Response

see file attached

This manuscript is a resubmission of an earlier submission. The following is a list of the peer review reports and author responses from that submission.

Round 1

Reviewer 1 Report

The manuscript aims to assess the level of implementation of one of the colonoscopy performance measures proposed by the ESGE, namely withdrawal time in centers from Europe. The authors analyzed the results from a questionnaire applyied between June 2016 and April 2018 to Europe-based colonoscopists. The main results indicated that sedation, in particular the use of midazolam and propofol was associated with longer withdrawal time. Moreover, the timing of the colonoscopy impacted the withdrawal time, with colonoscopies performed in the evening being associated with low withdrawal time.

There is evidence  indicating that lower adenoma detection rate is associated with colonoscopies performed in the evening. The results of this study indicate that one of the causes for this could be a shorter withdrawal time at the end of the day.

The mansucript is well structured, the methodology is sound and the conclusions are well formulated.

The study has however some limitations, which were however mostly stated by the authors.

There was no analysis of the influence of the day of the week regarding the withdrawal time. Moreover, the level of training of the colonoscopists involved in the study was not analyzed, nor was the type of institution. These issues could represent reasons for inadequate results.

All in all I consider that the manuscript could be published in its current state.

Author Response

Reviewer 1

Response to Reviewer 1

1.  Was this a prospective or retrospective study? It appears that data were prospectively entered but not reviewed for several years.

Data were entered prospectively and exploratory analysis performed.

2.  Please define the Boston Bowel prep scoring system (BBPS) if only in your supplemental information.

BBPS scoring system is shown in the questionnaire (attached).

BBPS reference inserted in Statistical Analysis (2.6)

3.  You look at a single quality measure: withdrawal time from the cecum. Adequacy of bowel prep, ADR, completeness of the exam, and adverse events are others. Were these measured and previously reported?

Measures (not ADR) were reported in:

Spada, C.; Koulaouzidis, A.; Hassan, C.; Amaro, P.; Agrawal, A.; Brink, L.; Fischbach, W.; Hünger, M.; Jover, R.; Kinnunen, U.; et al. Colonoscopy quality across Europe: a report of the European Colonoscopy Quality Investigation (ECQI) Group. Endosc Int Open 2021, 9, E1456-e1462, doi:10.1055/a-1486-6729

Cited as reference 2

Measures also reported in abstracts, see

https://ecqigroup.org/publications.html

4.  The reviewer is surprised that the institutions did not need to either a waiver or approval by Institutional Ethics Committees or IRBs given the fact that they are sharing data between institutions albeit most of which is de-identified.

We acknowledge that institutions and countries had varied protocols.

As stated: This survey recorded routine practice and ethical approval was not generally required, but participants were advised to obtain ethical approval, if appropriate, according to their local protocols.

5.  You analyzed 6445 completed data forms from 12 countries over 2 years suggesting that the numbers entered from the 51 participating centers are only a subset of the colonoscopies performed. Please comment.

In Limitations (4.2) we currently state “Selection bias is also possible as we have no control over which procedures practitioners choose to document.”

For practical reasons we did not specify which procedures should be documented and due to the additional time required to complete the questionnaire we could not request that all procedures were documented.

6.  Of the 6445 procedure forms, you report on 1150. What were the reasons for exclusion?

Please see new Figure 1. Flowchart indicating data inclusion/exclusion for analysis of withdrawal time (WT)

7.  You note that WT were longer in patients in whom there were endoscopic findings. What were the findings that did not require biopsy? Diverticulosis? Hemorhoids?

Text added:

Abnormal endoscopic findings included cancer (n=19), diverticulae (n=322), inflammation (n=46), polyps (n=52), haemorrhoids (n=44) and other findings (n=40).

8.  Please delineate the number of patients attributable to each hospital, academic center, practice or private institution.

Thank you for this suggestion. However, we consider that the current report on the number of procedures attributable to each institution type is adequate and a further breakdown would not add value.

Reviewer 2 Report

Thank you for submitting your manuscript to MDPI. Your research clearly took a lot of time and effort and it was an honor to review it. Please find my questions, comments, and suggestions below.

1. ln 94: the link to the supplementary material led to a 404 File Not Found webpage so I was unable to review the questionnaire. Could you please elaborate on how the six variables included in the multivariable stepwise model were arrived at? Were they the online variables in the questionnaire or were those six selected from a larger pool of variables using some technique and criteria?

2. ln 125. Please note and site the software that was used to complete the analysis

3. ln 125. Was a linear stepwise regression performed that kept the endpoint numeric?

4. ln 133. Did the stepwise logistic regression use AIC or p-value to determine which variables to add and/or remove from the model? What were the settings for the model? For example, if the stepwise logistic regression was performed using p-values as the criteria then there will be a p-value threshold for inclusion of a variable and one for exclusion. What were those thresholds?

5. ln 145. Only 17.8% of the completed forms met criteria for the WT analysis. Can you please elaborate on what the criteria was?

6. Of the 17.8% of forms that met criteria for WT analysis only 51% (or 9.2% overall) were able to be used for the stepwise logistics regression. Was multiple imputation considered? If so, why wasn't it used? In my opinion, the amount of missingness and the impact it could have on the results was under-represented in the limitation section.

7. Table 1. Please include a column indicating how much of each variable was missing.

8. Table 1/ln 125. Why was abnormal endoscopic finding not included in the multivariable model?

Comment. Since MDPI failed to provide a working link to your supplementary material I was unable to see what other variables would be available for the analysis. The following is just my general concern that may or may not be applicable to your research as a result.

Biased predictions are easy (easier) to come by when data that are used for selecting variables because of their predictive ability are also used to quantify how well the variables selected perform at predicting. Example: Say 500 observations were used to pick the 5 "best" variables at predicting an outcome and then those same 500 observations were used to quantify how well the 5 best variables did at predicting the outcome. It is a forgone conclusion that those 5 variables will perform really well. Because the reason those 5 variables were selected is that they performed well on those exact same observations. Ideally, a new set of patients (i.e. observations that were not used to select the variables) would be used to quantify how well those variables perform as predictors because the goal is to understand how well those variables might perform in future patients.

If the statistical methodology in the paper is similar to the one in my example, then please add that to the limitations section. The limitation being lack of validation methodology.

Author Response

1. The manuscript aims to assess the level of implementation of one of the colonoscopy performance measures proposed by the ESGE, namely withdrawal time in centers from Europe. The authors analyzed the results from a questionnaire applied between June 2016 and April 2018 to Europe-based colonoscopists. The main results indicated that sedation, in particular the use of midazolam and propofol was associated with longer withdrawal time. Moreover, the time of the colonoscopy impacted with withdrawal time, the colonoscopies performed in the evening being associated with low withdrawal time. 

2. There is evidence indicating that lower adenoma detection rate is associated with colonoscopies performed in the evening. The results of this study indicate that one of the causes for this could be a shorter withdrawal time at the end of the day. 

3. The manuscript is well structured, the methodology is sound and the conclusions are well formulated. 

4. The study has however some limitations, which were however mostly stated by the authors. There was no analysis of the influence of the day of the week regarding withdrawal time. Moreover, the level of training of the colonoscopists involved in the study was not analyzed nor was the type of institution. These issues could represent reasons for inadequate results. 

Reviewer 2 

Response to Reviewer 1 

5. The study has however some limitations, which were however mostly stated by the authors. There was no analysis of the influence of the day of the week regarding withdrawal time. Moreover, the level of training of the colonoscopists involved in the study was not analyzed nor was the type of institution. These issues could represent reasons for inadequate results. 

Text added to limitations: 

Additional variables that could have influenced the results include the day of the week colonoscopy was performed and the level of experience of the colonoscopist. These variables could be included in future questionnaires/analyses. 

6. All in all I consider that the manuscript could be published in its current state. 

Reviewer 3 Report

To the Authors:

  • Was this a prospective or retrospective study? It appears that data were prospectively entered but not reviewed for several years.
  • Please define the Boston Bowel prep scoring system (BBPS) if only in your supplemental information.
  • You look at a single quality measure; withdrawal time from the cecum. Adequacy of bowel prep, ADR, completeness of the exam, and adverse events are others. Were these measured and previously reported?
  • The reviewer is surprised that the institutions did not need either a waiver or approval by Institutional Ethics Committees or IRBs given the fact that they are sharing data between institutions albeit most of which is de-identified.
  • You analyzed 6445 completed data forms from 12 countries over 2 years suggesting that the numbers entered from the 51 participating centers are only a subset of the colonoscopies performed. Please comment.
  • Of the 6445 procedure forms, you report on 1150. What were the reasons for exclusion?
  • You note that WT were longer in patients in whom there were endoscopic findings. What were the findings that did not require biopsy? Diverticulosis? Hemorrhoids?
  • Please delineate the number of patients attributable to each hospital, academic center, practice or private institution.

Author Response

Thank you for submitting your manuscript to MDPI. Your research clearly took a lot of time and effort and it was an honor to review it. Please find my questions, comments, and suggestions below.

1.     ln 94: the link to the supplementary material led to a 404 File Not Found webpage so I was unable to review the questionnaire. Could you please elaborate on how the six variables included in the multivariable stepwise model were arrived at? Were they the online variables in the questionnaire or were those six selected from a larger pool of variables using some technique and criteria?

Procedure questionnaire is attached.

The variables selected for potential inclusion in the stepwise model were those which were considered, a priori, to be related to withdrawal time.

2.     ln 125. Please note and site the software that was used to complete the analysis

As stated: All analyses were performed using the statistical software package SAS version 9.4.

3.     In 125. Was a linear stepwise regression performed that kept the endpoint numeric?

The stepwise model for WT fitted a general linear model to the numeric variable. The stepwise model for each of the WT categories (e.g., WT < 6 minutes) fitted a binary logistic regression model.

4.     ln 133. Did the stepwise logistic regression use AIC or p-value to determine which variables to add and/or remove from the model? What were the settings for the model? For example, if the stepwise logistic regression was performed using p-values as the criteria then there will be a p-value threshold for inclusion of a variable and one for exclusion. What were those thresholds?

Variables were included in the stepwise model if the p value for entering the model were less than 0.05 and they are removed if their p value were greater than 0.10.

5.     ln 145. Only 17.8% of the completed forms met criteria for the WT analysis. Can you please elaborate on what the criteria was?

Please see new Figure 1. Flowchart indicating data inclusion/exclusion for analysis of withdrawal time (WT)

The procedure questionnaire was not developed to specifically address withdrawal time, rather all aspects of quality in colonoscopy.

6.     Of the 17.8% of forms that met criteria for WT analysis only 51% (or 9.2% overall) were able to be used for the stepwise logistics regression. Was multiple imputation considered? If so, why wasn't it used? In my opinion, the amount of missingness and the impact it could have on the results was under-represented in the limitation section.

Multiple imputation was not included in the Statistical Analysis Plan because these analyses are regarded as exploratory and doubts regarding whether the data were missing at random.

7.     Table 1. Please include a column indicating how much of each variable was missing.

Missing data included in Table 1.

8.     Table 1/ln 125. Why was abnormal endoscopic finding not included in the multivariable model?
Comment. Since MDPI failed to provide a working link to your supplementary material I was unable to see what other variables would be available for the analysis. The following is just my general concern that may or may not be applicable to your research as a result.

Biased predictions are easy (easier) to come by when data that are used for selecting variables because of their predictive ability are also used to quantify how well the variables selected perform at predicting. Example: Say 500 observations were used to pick the 5 "best" variables at predicting an outcome and then those same 500 observations were used to quantify how well the 5 best variables did at predicting the outcome. It is a forgone conclusion that those 5 variables will perform really well. Because the reason those 5 variables were selected is that they performed well on those exact same observations. Ideally, a new set of patients (i.e. observations that were not used to select the variables) would be used to quantify how well those variables perform as predictors because the goal is to understand how well those variables might perform in future patients.

If the statistical methodology in the paper is similar to the one in my example, then please add that to the limitations section. The limitation being lack of validation methodology.

This was an exploratory analysis, when variables for inclusion were decided it was not considered that abnormal endoscopic finding would be an influential factor. After analysis we decided to check, and confirmed that it was not.

Multiple imputation analysis performed to test sensitivity, and text added to limitations:

…, for example using training and validation subsets.